# Bayesian Hierarchical Community Discovery

**Charles Blundell**[*]
DeepMind Technologies
charles@deepmind.com

**Yee Whye Teh**
Department of Statistics,
University of Oxford
y.w.teh@stats.ox.ac.uk

## Abstract

We propose an efficient Bayesian nonparametric model for discovering hierarchical community structure in social networks. Our model is a tree-structured mixture of potentially exponentially many stochastic blockmodels. We describe a family of greedy agglomerative model selection algorithms that take just one pass through the data to learn a fully probabilistic, hierarchical community model. In the worst case, Our algorithms scale quadratically in the number of vertices of the network, but independent of the number of nested communities. In practice, the run time of our algorithms are two orders of magnitude faster than the Infinite Relational Model, achieving comparable or better accuracy.

## 1 Introduction

People often organise themselves into groups or communities. For example, friends may form cliques, scientists may have recurring collaborations, and politicians may form factions. Consequently the structure found in social networks is often studied by inferring these groups. Using community membership one may then make predictions about the presence or absence of unobserved connectivity in the social network. Sometimes these communities possess hierarchical structure. For example, within science, the community of physicists may be split into those working on various branches of physics, and each branch refined repeatedly until finally reaching the particular specialisation of an individual physicist.

Much previous work on social networks has focused on discovering flat community structure. The stochastic blockmodel [1] places each individual in a community according to the block structure of the social network's adjacency matrix, whilst the mixed membership stochastic blockmodel [2] extends the stochastic blockmodel to allow individuals to belong to several flat communities simultaneously. Both models require the number of flat communities to be known and are parametric methods.

Greedy hierarchical clustering has previously been applied directly to discovering hierarchical community structure [3]. These methods do not require the community structure to be flat or the number of communities to be known. Such schemes are often computationally efficient, scaling quadratically in the number of individuals for a dense network, or linearly in the number of edges for a sparse network [4]. These methods do not yield a full probabilistic account of the data, in terms of parameters and the discovered structure.

Several authors have also proposed Bayesian approaches to inferring community structure. The Infinite Relational Model (IRM; [5, 6, 7]) infers a flat community structure. The IRM has been extended to infer hierarchies [8], by augmenting it with a tree, but comes at considerable computational cost. [9] and [10] propose methods limited to hierarchies of depth two, whilst [11] propose methods limited to hierarchies of known depth.. The Mondrian process [12] propose a flexible prior on trees and a likelihood model for relational data. Current Bayesian nonparametric methods do not scale well to larger networks because the inference algorithms used make many small changes to the model.

---

[*]Part of the work was done whilst at the Gatsby Unit, University College London.

Such schemes can take a large number of iterations to converge on an adequate solution whilst each iteration often scales unfavourably in the number of communities or vertices.

We shall describe a greedy Bayesian hierarchical clustering method for discovering community structure in social networks. Our work builds upon Bayesian approaches to greedy hierarchical clustering [13, 14] extending these approaches to relational data. We call our method Bayesian Hierarchical Community Discovery (BHCD). BHCD produces good results two orders of magnitude faster than a single iteration of the IRM, and its worst case run-time is quadratic in the number of vertices of the graph and independent of the number of communities.

The remainder of the paper is organised as follows. Section 2 describes the stochastic blockmodel. In Section 3 we introduce our model as a hierarchical mixture of stochastic blockmodels. In Section 4 we describe an efficient scheme for inferring hierarchical community structure with our model. Section 5 demonstrates BHCD on several data sets. We conclude with a brief discussion in Section 6

## 2    Stochastic Blockmodels

A stochastic blockmodel [1] consists of a partition, $\phi$, of vertices $V$ and for each pair of clusters $p$ and $q$ in $\phi$, a parameter, $\theta_{pq}$, giving the probability of a presence or absence of an edge between nodes of the clusters. Suppose $V = \{a, b, c, d\}$, then one way to partition the vertices would be to form clusters $ab$, $c$ and $d$, which we shall write as $\phi = ab|c|d$, where $|$ denotes a split between clusters. The probability of an adjacency matrix, $\mathcal{D}$, given a stochastic blockmodel, is as follows:

$$P(\mathcal{D}|\phi, \{\theta_{pq}\}_{p,q\in\phi}) = \prod_{p,q\in\phi} \theta_{pq}^{n^1_{pq}}(1 - \theta_{pq})^{n^0_{pq}} \tag{1}$$

where $n^1_{pq}$ is the total number of edges in $\mathcal{D}$ between the vertices in clusters $p$ and $q$, and $n^0_{pq}$ is the total number of observed absent edges in $\mathcal{D}$ between the vertices in clusters $p$ and $q$.

When modelling communities, we expect the edge appearance probabilities within a cluster to be different to those between different clusters. Hence we place a different prior on each of these cases. Similar approaches have been taken to adapt the IRM to community detection [7], where non-conjugate priors were used at increased computational cost. In the interest of computational efficiency, our model will instead use conjugate priors but with differing hyperparameters. $\theta_{pp}$ will have a Beta$(\alpha, \beta)$ prior and $\theta_{pq}, p \neq q$, will have a Beta$(\delta, \lambda)$ prior. The hyperparameters are picked such that $\alpha > \beta$ and $\delta < \lambda$, which correspond to a prior belief of a higher density of edges within a community than across communities. Integrating out the edge appearance parameters, we obtain the following likelihood of a particular partition $\phi$:

$$P(\mathcal{D}|\phi) = \prod_{p\in\phi} \frac{\mathrm{B}(\alpha + n^1_{pp}, \beta + n^0_{pp})}{\mathrm{B}(\alpha, \beta)} \prod_{\substack{p,q\in\phi \\ p\neq q}} \frac{\mathrm{B}(\delta + n^1_{pq}, \lambda + n^0_{pq})}{\mathrm{B}(\delta, \lambda)} \tag{2}$$

where $\mathrm{B}(\cdot, \cdot)$ is the Beta function. We may generalise this to use any exponential family:

$$p(\mathcal{D}|\phi) = \prod_{p\in\phi} f(\sigma_{pp}) \prod_{p,q\in\phi,\, p\neq q} g(\sigma_{pq}) \tag{3}$$

where $f(\cdot)$ is the marginal likelihood of the on-diagonal blocks, and $g(\cdot)$ is the marginal likelihood of the off-diagonal blocks. We use $\sigma_{pq}$ to denote the sufficient statistics from a $(p, q)$-block of the adjacency matrix: all of the elements whose row indices are in cluster $p$ and whose column indices are in cluster $q$. For the remainder of the paper, we shall focus on the beta-Bernoulli case given in (2) for concreteness. i.e., $\sigma_{pq} = (n^1_{pq}, n^0_{pq})$, with $f(x, y) = \frac{B(\alpha+x, \beta+y)}{B(\alpha, \beta)}$ and $g(x, y) = \frac{B(\delta+x, \lambda+y)}{B(\delta, \lambda)}$. For clarity of exposition, we shall focus on modelling undirected or symmetric networks with no self-edges, so $\sigma_{pq} = \sigma_{qp}$ and $\sigma_{\{x\}\{x\}} = 0$ for each vertex $x$, but in general this restriction is not necessary.

One approach to inferring $\phi$ is to fix the number of communities $K$ then use maximum likelihood estimation or Bayesian inference to assign vertices to each of the communities [1, 15]. Another approach is to use variational Bayes, combined with an upper bound on the number of communities, to determine the number of communities and community assignments [16].

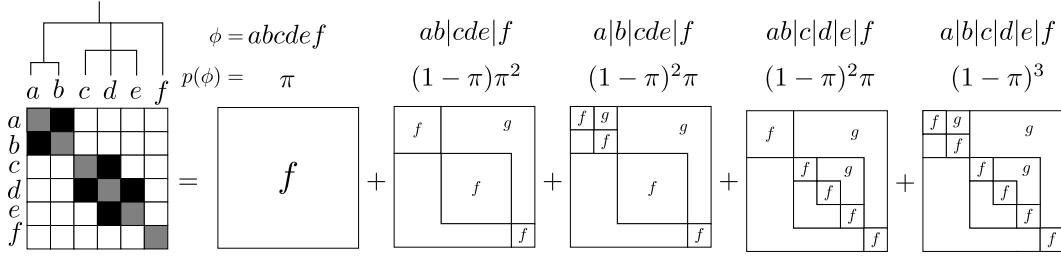

Figure 1: Hierarchical decomposition of the adjacency matrix into tree-consistent partitions. Black squares indicated edge presence, white squares indicate edge absence, grey squares are unobserved.

## 3  Bayesian Hierarchical Communities

In this section we shall develop a Bayesian nonparametric approach to community discovery. Our model organises the communities into a nested hierarchy $T$, with all vertices in one community at the root and singleton vertices at the leaves. Each vertex belongs to all communities along the path from the root to the leaf containing it. We describe the probabilistic model relating the hierarchy of communities to the observed network connectivity data here, whilst in the next section we will develop a greedy model selection procedure for learning the hierarchy $T$ from data.

We begin with the marginal probability of the adjacency matrix $\mathcal{D}$ under a stochastic blockmodel:

$$p(\mathcal{D}) = \sum_{\phi} p(\phi)p(\mathcal{D}|\phi) \tag{4}$$

If the Chinese restaurant process (CRP) is used as the prior on partitions $p(\phi)$, then (4) corresponds to the marginal likelihood of the IRM. Computing (4) typically requires an approximation: the space of partitions $\phi$ is large and so the number of partitions in the above sum grows at least exponentially in the number of vertices.

We shall take a different approach: we use a tree to define a prior on partitions, where only partitions that are consistent with the tree are included in the sum. This allows us to evaluate (4) exactly. The tree will represent the hierarchical community structure discovered in the network. Each internal node of the tree corresponds to a community and the leaves of the tree are the vertices of the adjacency matrix, $\mathcal{D}$. Figure 1 shows how a tree defines a collection of partitions for inclusion in the sum in (4). The adjacency matrix on the left is explained by our model, conditioned upon the tree on the upper left, by its five tree-consistent partitions. Various blocks within the adjacency matrix are explained either by the on-diagonal model $f$ or the off-diagonal model $g$, according to each partition. Note that the block structure of the off-diagonal model depends on the structure of the tree $T$, not just on the partition $\phi$. The model always includes the trivial partition of all vertices in a single community and also the singleton partition of all vertices in separate communities.

More precisely, we shall denote trees as a nested collection of sets of vertices. For example, the tree in Figure 1 is $T = \{\{a, b\}, \{c, d, e\}, f\}$. The set of of partitions consistent with a tree $T$ may be expressed formally as in [14]:

$$\boldsymbol{\Phi}(T) = \{\text{leaves}(T)\} \cup \{\phi_1| \ldots |\phi_{n_T} : \phi_i \in \boldsymbol{\Phi}(T_i), T_i \in \text{ch}(T)\} \tag{5}$$

where leaves$(T)$ are the leaves of the tree $T$, ch$(T)$ are its children, and so $T_i$ is the $i$th subtree of tree $T$. The marginal likelihood of the tree $T$ can be written as:

$$p(\mathcal{D}|T) = p(\mathcal{D}_{TT}|T) = \sum_{\phi} p(\phi|T)p(\mathcal{D}_{TT}|\phi, T) \tag{6}$$

where the notation $\mathcal{D}_{TT}$ is short for $\mathcal{D}_{\text{leaves}(T),\text{leaves}(T)}$, the block of $\mathcal{D}$ whose rows and columns correspond to the leaves of $T$.

Our prior on partitions $p(\phi|T)$ is motivated by the following generative process: Begin at the root of the tree, $S = T$. With probability $\pi_S$, stop and generate $\mathcal{D}_{SS}$ according to the on-diagonal model $f$. Otherwise, with probability $1 - \pi_S$, generate all inter-cluster edges between the children of the current node according to $g$, and recurse on each child of the current tree $S$. The resulting prior on

tree-consistent partitions $p(\phi|T)$ factorises as:

$$p(\phi|T) = \prod_{S \in \mathrm{ancestor}_T(\phi)} (1 - \pi_S) \prod_{S \in \mathrm{subtree}_T(\phi)} \pi_S \tag{7}$$

where $\mathrm{subtree}_T(\phi)$ are the subtrees in $T$ corresponding to the clusters in partition $\phi$ and $\mathrm{ancestor}_T(\phi)$ are the ancestors of trees in $\mathrm{subtree}_T(\phi)$. The prior probability of partitions not consistent with $T$ is zero. Following [14], we define $\pi_S = 1 - (1 - \gamma)^{|\mathrm{ch}(S)|}$, where $\gamma$ is a parameter of the model. This choice of $\pi_S$ gives higher likelihood to non-binary trees over cascading binary trees when the data has no hierarchical structure [14]. Similarly, the likelihood of each partition $p(\mathcal{D}|\phi, T)$ factorises as:

$$p(\mathcal{D}_{TT}|\phi, T) = \prod_{S \in \mathrm{ancestor}_T(\phi)} g\left(\sigma_{SS}^{\neg \mathrm{ch}}\right) \prod_{S \in \mathrm{subtree}_T(\phi)} f(\sigma_{SS}) \tag{8}$$

where $\sigma_{SS}$ are the sufficient statistics of the adjacency matrix $\mathcal{D}$ among the leaves of tree $S$, and $\sigma_{SS}^{\neg \mathrm{ch}}$ are the sufficient statistics of the edges between different children of $S$:

$$\sigma_{SS}^{\neg \mathrm{ch}} = \sigma_{SS} - \sum_{C \in \mathrm{ch}(S)} \sigma_{CC} \tag{9}$$

The set of tree consistent partitions given in (5) has at most $O(2^n)$ partitions, for $n$ vertices. However due to the structure of the prior on partitions (7) and the block model (8), the marginal likelihood (6) may be calculated by dynamic programming, in $O(n + m)$ time where $n$ is the number of vertices and $m$ is the number of edges. Combining (7) and (8) and expanding (6) by breadth-first traversal of the tree, yields the following recursion for the marginal likelihood of the generative process given at the beginning of the section:

$$p(\mathcal{D}_{TT}|T) = \pi_T f(\sigma_{TT}) + (1 - \pi_T) g\left(\sigma_{TT}^{\neg \mathrm{ch}}\right) \prod_{C \in \mathrm{ch}(T)} p(\mathcal{D}_{CC}|C) \tag{10}$$

## 4 Agglomerative Model Selection

In this section we describe how to learn the hierarchy of communities $T$. The problem is treated as one of greedy model selection: each tree $T$ is a different model, and we wish to find the model that best explains the data. The tree is built in a bottom-up greedy agglomerative fashion, starting from a forest consisting of $n$ trivial trees, each corresponding to exactly one vertex. Each iteration then merges two of the trees in the forest. At each iteration, each vertex in the network is a leaf of exactly one tree in the forest. The algorithm finishes when just one tree remains. We define the likelihood of the forest $F$ as the probability of data described by each tree in the forest times that for the data corresponding to edges between different trees:

$$p(\mathcal{D}|F) = g(\sigma_{FF}^{\neg \mathrm{ch}}) \prod_{T \in F} p(\mathcal{D}_{TT}|T) \tag{11}$$

where $\sigma_{FF}^{\neg \mathrm{ch}}$ are the sufficient statistics of the edges between different trees in the forest.

The initial forest, $F^{(0)}$, consists a singleton tree for each vertex of the network. At each iteration a pair of trees in the forest $F$ is chosen to be merged, resulting in forest $F^\star$. Which pair of tree to merge, and how to merge these trees, is determined by considering which pair and type of merger yields the largest Bayes factor improvement over the current model. If the trees $I$ and $J$ are merged to form the tree $M$, then the Bayes factor score is:

$$\mathrm{SCORE}(M; I, J) = \frac{p(\mathcal{D}_{MM}|F^\star)}{p(\mathcal{D}_{MM}|F)} = \frac{p(\mathcal{D}_{MM}|M)}{p(\mathcal{D}_{II}|I) p(\mathcal{D}_{JJ}|J) g(\sigma_{IJ})} \tag{12}$$

where $p(\mathcal{D}_{MM}|M)$, $p(\mathcal{D}_{II}|I)$ and $p(\mathcal{D}_{JJ}|J)$ are given by (10) and $\sigma_{IJ}$ are the sufficient statistics of the edges connecting $\mathrm{leaves}(I)$ and $\mathrm{leaves}(J)$. Note that the Bayes factor score is based on data local to the merge—i.e., by considering the probability of the connectivity data only among the leaves of the newly merged tree. This permits efficient local computations and makes the assumption that local community structure should depend only on the local connectivity structure.

We consider three possible mergers of two trees $I$ and $J$ into $M$. See Figure 2, where for concreteness we take $I = \{T_a, T_b, T_c\}$ and $J = \{T_d, T_e\}$ where $T_a, T_b, T_c, T_d, T_e$ are subtrees. $M$ may be

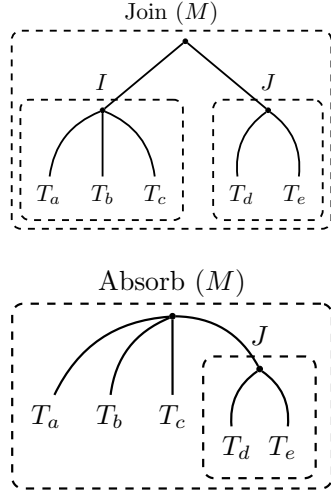

Join $(M)$

$T_a$ $T_b$ $T_c$ | $T_d$ $T_e$

Absorb $(M)$

$T_a$ $T_b$ $T_c$ $T_d$ $T_e$

Figure 2: Different merge operations.

1: Initialise $F, \{p_I, \sigma_{II}^{\neg\text{ch}}\}_{I \in F}, \{\sigma_{IJ}\}_{I,J \in F}$.
2: **for each unique pair** $I, J \in F$ **do**
3:    Let $M := \text{MERGE}(I; J)$, compute $p_M$ and $\text{SCORE}(M; I, J)$, and add $M$ to the heap.
4: **end for**
5: **while** heap is not empty **do**
6:    Pop $I = \text{MERGE}(X; Y)$ off the top of heap.
7:    **if** $X \in F$ and $Y \in F$ **then**
8:       $F \leftarrow (F \setminus \{X, Y\}) \cup \{I\}$.
9:       **for each tree** $J \in F \setminus \{I\}$, **do**
10:          Compute $\sigma_{IJ}, \sigma_{MM}$, and $\sigma_{MM}^{\neg\text{ch}}$ using (13).
11:          Let $M := \text{MERGE}(I; J)$, compute $p_M$ and $\text{SCORE}(M; I, J)$, and add $M$ to heap.
12:       **end for**
13:    **end if**
14: **end while**
15: **return** the only tree in $F$

Algorithm 1: Bayesian hierarchical community discovery.

formed by joining $I$ and $J$ together using a new node, giving $M = \{I, J\}$. Alternatively $M$ may be formed by absorbing $J$ as a child of $I$, yielding $M = \{J\} \cup \text{ch}(I)$, or vice versa, $M = \{I\} \cup \text{ch}(J)$.

The algorithm for finding $T$ is described in Algorithm 1. The algorithm maintains a forest $F$ of trees, the likelihood $p_I = p(\mathcal{D}_{II}|I)$ of each tree $I \in F$ according to (10), and the sufficient statistics $\{\sigma_{II}^{\neg\text{ch}}\}_{I \in F}$, $\{\sigma_{IJ}\}_{I,J \in F}$ needed for efficient computation. It also maintains a heap of potential merges ordered by the SCORE (12), used for determining the ordering of merges. At each iteration, the best potential merge, say of trees $X$ and $Y$ resulting in tree $I$, is picked off the heap. If either $X$ or $Y$ is not in $F$, this means that the tree has been used in a previous merge, so that the potential merge is discarded and the next potential merge is considered. After a successful merge, the sufficient statistics associated with the new tree $I$ are computed using the previously computed ones:

$$\sigma_{IJ} = \sigma_{XJ} + \sigma_{YJ} \qquad \text{for } J \in F, J \neq I.$$
$$\sigma_{MM} = \sigma_{II} + \sigma_{JJ} + \sigma_{IJ}$$
$$\sigma_{MM}^{\neg\text{ch}} = \begin{cases} \sigma_{IJ} & \text{if } M \text{ is formed by joining } I \text{ and } J, \\ \sigma_{II}^{\neg\text{ch}} + \sigma_{IJ} & \text{if } M \text{ is formed by } J \text{ absorbed into } I, \\ \sigma_{JJ}^{\neg\text{ch}} + \sigma_{IJ} & \text{if } M \text{ is formed by } I \text{ absorbed into } J. \end{cases} \qquad (13)$$

These sufficient statistics are computed based on previous cached values, allowing each inner loop of the algorithm to take $O(1)$ time. Finally, potential mergers of $I$ with other trees $J$ in the forest are considered and added onto the heap. In the algorithm, $\text{MERGE}(I; J)$ denotes the best of the three possible merges of $I$ and $J$.

Algorithm 1 is structurally the same as that in [14], and so has time complexity in $O(n^2 \log(n))$. The difference is that additional care is needed to cache the sufficient statistics allowing for $O(1)$ computation per inner loop of the algorithm. We shall refer to Algorithm 1 as BHCD.

## 4.1 Variations

BHCD will consider merging trees that have no edges between them if the merge score (12) is high enough. This does not seem to be a reasonable behaviour as communities that are completely disconnected should not be merged. We can alter BHCD by simply prohibiting such merges between trees that have no edges between them. The resulting algorithm we call BHCD sparse, as it only needs to perform computations on the parts of the network with edges present. Empirically, we have found that it produces better results than BHCD and runs faster for sparse networks, although in the worst case it has the same time complexity $O(n^2 \log n)$ as BHCD.

As BHCD runs, several merges can have the same score. In particular, at the first iteration all pairs of vertices connected by an edge have the same score. In such situations, we break the ties at random. Different tie breaks can yield different results and so different runs on the same data may yield

different trees. Where we want a single tree, we use $R$ ($R = 50$ in experiments) restarts and pick the tree with the highest likelihood according to (10). Where we wish to make predictions, we will construct predictive probabilities (see next section) by averaging all $R$ trees.

## 4.2 Predictions

For link prediction, we wish to obtain the predictive distribution of a previously unobserved element of the adjacency matrix. This is easily achieved by traversing one path of the tree from the root towards the leaves, hence the computational complexity is linear in the depth of the tree. In particular, suppose we wish to predict the edge between $x$ and $y$, $\mathcal{D}_{xy}$, given the observed edges $\mathcal{D}$, then the predictive distribution can be computed recursively starting with $S = T$:

$$p(\mathcal{D}_{xy}|\mathcal{D}_{SS}, S) = r_S f(\mathcal{D}_{xy}|\sigma_{SS}) + (1 - r_S) \begin{cases} p(\mathcal{D}_{xy}|\mathcal{D}_{CC}, C) & \text{if } \exists C \in \text{ch}(S) : x, y \in \text{leaves}(C), \\ g(\mathcal{D}_{xy}|\sigma_{SS}^{\neg\text{ch}}) & \text{if } \forall C \in \text{ch}(S) : x, y \notin \text{leaves}(C). \end{cases}$$

$$r_S = \frac{\pi_S f(\sigma_{SS})}{p(\mathcal{D}_{SS}|S)} \tag{14}$$

where $r_S$ is the probability that the cluster consisting of leaves$(S)$ is present if the cluster corresponding to its parent is not present, given the data $\mathcal{D}$ and the tree $T$. The predictive distribution is a mixture of a number of on-diagonal posterior $f$ terms (weighted by the responsibility $r_T$), and finally an off-diagonal posterior $g$ term. Hence the computational complexity is $\Theta(n)$.

# 5 Experiments

We now demonstrate BHCD on three data sets. Firstly we examine qualitative performance on Sampson's monastery network. Then we demonstrate the speed and accuracy of our method on a subset of the NIPS 1–17 co-authorship network compared to IRM—one of the fastest Bayesian nonparametric models for these data. Finally we show hierarchical structure found examining the full NIPS 1–17 co-authorship network. In our experiments we set the model hyperparameters $\alpha = \delta = 1.0$, $\beta = \lambda = 0.2$, and $\gamma = 0.4$ which we found to work well. In the first two experiments we shall compare four variations of BHCD: BHCD, BHCD sparse, BHCD restricted to binary trees, and BHCD sparse restricted to binary trees. Binary-only variations of BHCD only consider joins, not absorptions, and so may run faster. They also tend to produce better predictive results as they average over a larger number of partitions. But, as we shall see below, the hierarchies found can be more difficult to interpret than the non-binary hierarchies found.

**Sampson's Monastery Network** Figure 3 shows the result of running six variants of BHCD on time four of Sampson's monastery network [17]. Sampson observed the monastery at five times—time four is the most interesting time as it was before four of the monks were expelled. We treated positive affiliations as edges, and negative affiliations as observed absent edges, and unknown affiliation as missing data. [17], using a variety of methods, found four flat groups, shown at the top of Figure 3: Young Turks (Albert, Boniface, Gregory, Hugh, John Bosco, Mark, Winfrid), Loyal Opposition (Ambrose, Berthold, Bonaventure, Louis, Peter), Outcasts (Basil, Elias, Simplicius), and Interstitial group (Amand, Ramuald, Victor).

As can be seen in Figure 3, most BHCD variants find clear block diagonal structure in the adjacency matrix. The binary versions find similar structure to the non-binary versions, up to permutations of the children of the non-binary trees. BHCD global is lead astray by out of date scores on its heap and so finds less coherent structure. The log likelihoods of the trees in Figure 3 are $-6.62$ (BHCD) and $-22.80$ (BHCD sparse). Whilst the log likelihoods of the binary trees in Figure 3 are $-8.32$ (BHCD binary) and $-24.68$ (BHCD sparse binary). BHCD finds the most likely tree, and rose trees typically better explain the data than binary trees.

BHCD finds the Young Turks and Loyal Opposition groups and chooses to merge some members of the Interstitial group into the Loyal Opposition and Amand into the Outcasts. Mark, however, is placed in a separate community: although Mark has a positive affiliation with Gregory, Mark also has a negative affiliation with John Bosco and so BHCD elects to create a new community to account for this discrepancy.

**NIPS-234** Next we applied BHCD to a subset of the NIPS co-authorship dataset [19]. We compared its predictive performance to both IRM using MCMC and also inference in the IRM using greedy

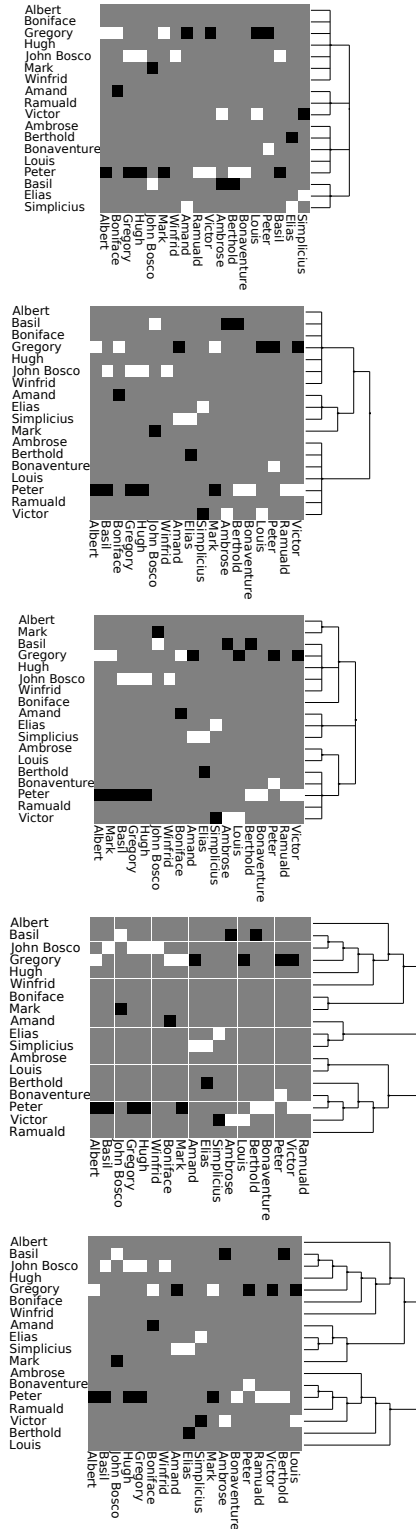

Figure 3: Sampson's monastery network. White indicates a positive affiliation, black negative, whilst grey indicates unknown. From top to bottom: Sampson's clustering, BHCD, BHCD-sparse, BHCD with binary trees, BHCD-sparse-binary.

| Method | Time complexity |
|---|---|
| IRM (naïve) | $O(n^2 K^2 IR)$ |
| IRM (sparse) | $O(m K^2 IR)$ |
| LFRM [18] | $O(n^2 F^2 IR)$ |
| IMMM [9] | $O(n^2 K^2 IR)$ |
| ILA [10] | $O(n^2(F + K^2)IR)$ |
| [8] | $O(n^2 K^2 IR)$ |
| BHCD | $O(n^2 \log(n)R)$ |

Table 1: Time complexities of different methods. $n$ = # vertices, $m$ = # edges, $K$ = # communities, $F$ = # latent factors, $I$ = # iterations per restart, $R$ = # restarts.

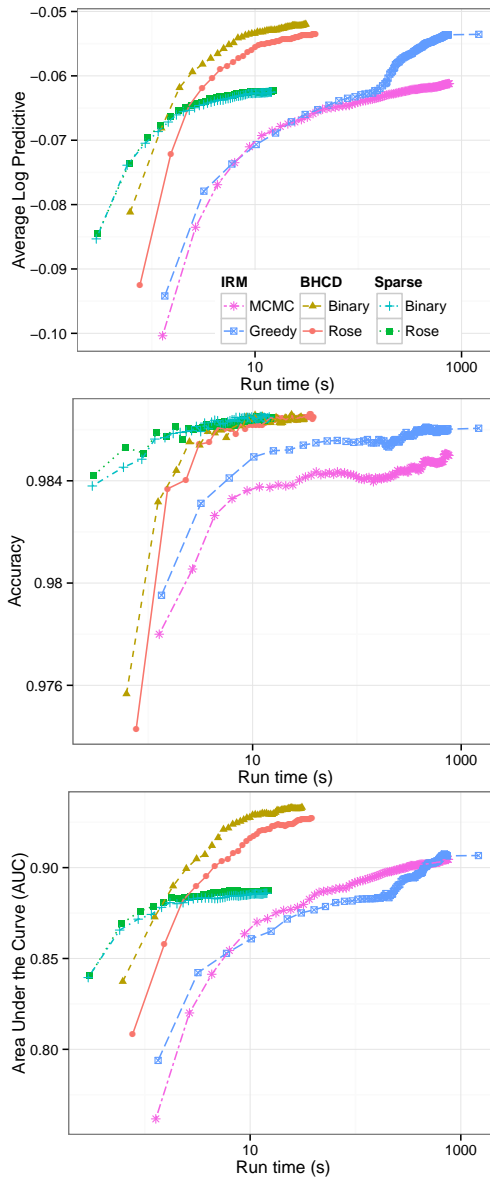

Figure 4: NIPS-234 comparison using log predictive, accuracy and AUC, averaged across 10 cross-validation folds.

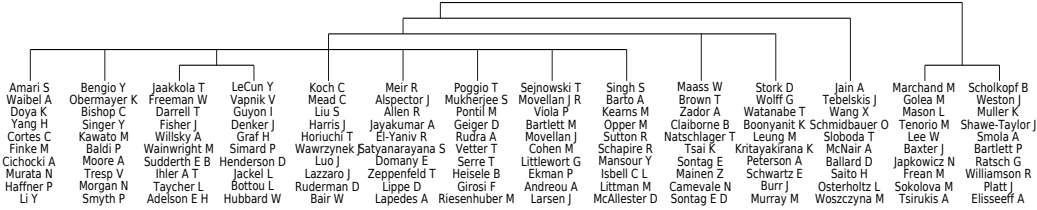

Figure 5: Clusters of authors found in NIPS 1–17. Top 10 most most collaborating authors shown for all clusters with more than 15 vertices.

search, using the publicly available C implementation[20]. Our implementation of BHCD is also in C. As can be seen from Table 1, BHCD has significantly lower computational complexity than other Bayesian nonparametric methods even than those inferring flat hierarchies. This is because it is a simpler model and uses a simpler inference method—thus we do not expect it to yield better predictive results, but instead to get good results quickly. Unlike the other listed methods, BHCD's worst case complexity does not depend upon the number of communities, and BHCD always terminates after a fixed number of steps so has no $I$ factor. This latter factor, $I$, corresponds to the number of MCMC steps or the number of greedy search steps, may be large and may need to scale as the number of vertices increases.

Following [18, 10] we restricted the network to the 234 most connected individuals. Figure 4 shows the average log predictive probability of held out data, accuracy and Area under the receiver operating Curve (AUC) over time for both BHCD and IRM. For the IRM, each point represents a single Gibbs step (for MCMC) or a search step (for greedy search). For BHCD, each point represents a complete run of the inference algorithm. BHCD is able to make reasonable predictions before the IRM has completed a single Gibbs scan. We used the same 10 cross-validation folds as used in [10] and so our results are quantitatively comparable to their results for the Latent Factor Relational Model (LFRM [18]) and their model, the Infinite Latent Attributes model (ILA). BHCD performs similarly to LFRM, worse than ILA, and better IRM. After about 10 seconds, the sparse variants of BHCD make as good predictions on NIPS-234 as the IRM after about 1000 seconds. Notably the sparse variations are faster than the non-sparse variants of BHCD, as the NIPS co-authorship network is sparse.

**Full NIPS** The full NIPS dataset has 2864 vertices and 9466 edges. Figure 5 shows part of the hierarchy discovered by BHCD. The full inferred hierarchy is large, having 646 internal nodes. We cut the tree and retained the top portion of the hierarchy, shown above the clusters. We merged all the leaves of a subtree $T$ into a flat cluster when $r_T \prod_{A \in \text{ancestor}_T} (1 - r_A) > 0.5$ where $r_T$ is given in (14). This quantity corresponds to the probability of picking that particular subtree in the predictive distribution. Amongst these clusters we included only those with at least 15 members in Figure 5. We include hierarchies with a lower cut-off in the supplementary.

## 6 Discussion and Future Work

We proposed an efficient Bayesian procedure for discovering hierarchical communities in social networks. Experimentally our procedure discovers reasonable hierarchies and is able to make predictions about two orders of magnitude faster than one of the fastest existing Bayesian nonparametric schemes, whilst attaining comparable performance. Our inference procedure scales as $O(n^2 \log n)$ through a novel caching scheme, where $n$ is the number of vertices, making the procedure suitable for large dense networks. However our likelihood can be computed in $O(n + m)$ time, where $m$ are the number of edges. This disparity between inference and likelihood suggests that in future it may be possible to improve the scalability of the model on sparse networks, where $m \ll n^2$. Another way to scale up the model would be to investigate parameterising the network using the sufficient statistics of triangles, instead of edges as in [21]. Others [7] have found that non-conjugate likelihoods can yield improved predictions—thus adapting our scheme to work with non-conjugate likelihoods and doing hyperparameter inference could also be fruitful next steps.

**Acknowledgements** We thank the Gatsby Charitable Foundation for generous funding.

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
