[Reviews · NeurIPS 2013]

Submitted by Assigned_Reviewer_2

The authors present a hierarchical extension of the IRM for network modelling using the key ideas from the Bayesian rose tree paper: 1) that the hierarchy is used to specify a mixture over consistent partitions of the nodes 2) that this hierarchy can be learnt using an efficient greedy agglomerative procedure. Qualitative results on the Sampson's monks dataset, and full NIPS dataset, and quantitative results on the NIPS-234 dataset are presented. The proposed inference is computational much cheaper than the IRM, whilst obtaining similar predictive performance.

Quality/clarity. The paper is very well written and the exposition of the key ideas is clear.

Originality. The main ideas are taken from the Bayesian rose tree work, but there was some work to do in adapting these ideas to the relational data setting, particularly in maintaining the same computational complexity.

Significance. Scaling BNP models to, especially network models, is an important challenge if these methods are to be used by practioners. While the n^2 scaling of the basic algorithm may prove prohibitive for larger networks, the sparse version should presumably remain feasible. It would have been nice to see some timings on larger real world networks. NIPS-234 takes only tens of seconds: how long does full NIPS take? Gene networks would typically have tens of thousands of nodes, and real social networks are much larger still: how would inference scale?

In addition it would be nice to have some feel how close to the optimal tree the greedy method gets. This could be investigated at the largest scale that you can still find the optimal tree by exhaustive search. Is it possible to given any theoretical support to the method? e.g. in terms of submodularity?

Minor corrections:
Line 48 page 1: whilst... but
pair of treeS
A fruitful next steps
Summary: While the ideas presented here are familiar from the BRT work the contribution is still significant and this paper should be accepted.

Submitted by Assigned_Reviewer_4

The authors propose a generalization of the Bayesian nonparamtric stochastic block model to the hierarchical setting, in order to infer a hierarchy rather than a flat clustering of entities when conditioned on observed links between entities. They propose a learning algorithm based on agglomerative clustering to select a tree which represents valid possible partitions of the entities. They then present empirical results from two corpuses of social connectivity data and conclude that their method is both faster and more accurate than the Infinite Relational Model (IRM), a widely-used model for tasks of this kind.

I believe this paper tackles an interesting issue that a significant fraction of the NIPS audience will be interested in. On the technical side, Bayesian nonparametric methods, while attractive from a conceptual point of view, are often quite slow in practice. On the proble domain side, social scientsits are increasingly using sophisticated Bayesian models for analyzing data and so a hierarchical extension to a popular network-discovery model should be appreciated. This paper touches on both those issues.

The key insight is that by restricting allowed partitions to conform to a tree structure, the marginal likelihood of the data can be computed efficiently. The IRM must essentially consider the exponential number of possible partitions of the data. However, I was left somewhat confused on how the learning procedure of the trees relates to the generative story behind the Bayesian model. Since it is greedy, I do not think it can be seen as performing true posterior inference over a distribution of trees or finding some kind of point MAP or ML estimate. But I may have misunderstood. I would recommend that the authors try to improve the clarity of Section 4.

I found the results to be impressive. In terms of several performance measures (AUC, predictive llh, accuracy), the proposed method performed comparably or slightly better than the IRM, the closest competitor. While the asymptotic improvements in these scores are perhaps not that impresive in absolute terms, the claimed runtime is orders of magnitude faster than IRM inference.

I think the overall clarity and writing quality could be improved. First, there are a series of distracting spelling and grammar errors (eg line 126). Second, the prose is sometimes hard to understand. Line 47-50 in the introduction has some vague phrasing (what are 'local changes' in this setting? What are the 'iterations' referring to? Iterations of an MCMC algorithm?), for instance. The authors do not clearly describe the content of the datasets they train on (Monastery networks and NIPS authorship), which could confuse readers who are not already familiar with them. I occasionally found the math somewhat obtuse and unnecessary, such as Eq. 5. I also found Figure 3 confusing. Being unfamiliar with the monastery dataset, it is visually unclear which method is performing the best. Perhaps a dataset that more readers will be familiar with would be helpful. Section 4, which presents the core novel contribution of the paper, required several readings to understanding.
If these issues could be remedied, however, I believe the proposed approach has enough novelty and the results are strong enough to warrant publication.
The authors propose a novel model of graph edge structure that can be seen as a Bayesian version of agglomerative clustering inspired by the IRM. I found the core learning algorithm to be insightful and the experimental results to be strong, but the quality of writing in need of improvement.
Summary: The authors propose a novel model of graph edge structure that can be seen as a Bayesian version of agglomerative clustering inspired by the IRM. I found the core learning algorithm to be insightful and the experimental results to be strong, but the quality of writing in need of improvement.

Submitted by Assigned_Reviewer_6

Well written paper with good spelling, grammar, correct citations, etc. Well above average here. Though try not to begin a sentence with a citation in the NIPS style. Theory is succinct in parts, for instance using f() and g() throughout the paper.
The paper looks at stochastic block models (SBM) and extends the simple single table instance of an IRM by adding the theory of Bayesian Rose trees. This allows the CRP formulation of IRM to be replaced by the simpler eq (7). The factoring over subtrees follows the work on learning Bayesian network structures, e.g., Koivisto and Sood JMLR 2004 section 3, where the posterior factorises neatly based on the class of *constrainted* structures chosen and thus greedy search, marginal likelihoods, handling of sufficient statistics, link prediction, etc., can all be neatly done.
Evaluation for SBM is often done against "gold standard" data where the partition structure is known. Scores like NMI and other clustering metrics can be used for this where you compare against the known clusterings. You report using AUC and I had to think for a while how this could be done, since am familar with NMI evaluations for SBM. I presume you delete a few links and then guess if they should be added back. Is this right? Anyway, badly needs an explanation. Recent evaluations of SBM run on bigger data sets, bigger than your full NIPS, so the small evaluation shown in Figure 4 is good but should be presented for a larger dataset too. Its not good enough just to compare against Bayesian non-parametric methods as there are many published semi-parametric methods, some you mention.
The contribution of this algorithm is the agglomerative method and Table 1 sums it up well.
Summary: The paper extends the simple single table instance of an IRM by adding the theory of Bayesian Rose trees. Good clear theory and algorithms, though a simple concept and algorithms. The experimental work is illustrative rather than comparative so not really adequate.

Submitted by Assigned_Reviewer_7

Proposes a agglomerative approach to scalably clustering networks into hierarchical communities. The prior over (possibly non-binary) trees, and algorithm structure, are essentially unchanged from prior work on Bayesian rose trees [12]. The likelihood is modified in a straightforward way to handle networks, and a bit more work allows sufficient statistics to still be cached efficiently. The presentation is very clear and experiments compare favorably to a baseline model with flat community structure, but comparisons to other hierarchical network models are lacking.

SIGNIFICANCE & RELATED WORK:
The paper is generality well written and seems to be technically sound. The presentation in Sec. 3 is particularly dense, I wonder if there might be a simpler way of notating and explaining (9) and the parts which build on it.

Since comparisons are limited to models that do not incorporate hierarchy in their latent structure, experiments mostly show how modeling hierarchy is beneficial for relational datasets, but not whether they are better than other hierarchical methods that currently exist. For example, there are at least two papers that model the underlying latent community structure in a hierarchical fashion. The "multiscale community blockmodel for Network exploration" (Ho et. al 2012) utilizes a nested CRP and the "Mondrian Process" (Roy et. al 2009) defines a novel stochastic process that results in hierarchical structures over K-d trees as its prior. The paper fails to mention these two Bayesian nonparametric models which potentially are capable of discovering a more refined latent structure.

ORIGINALITY & CLARITY:
The model considered here seems to be a Bayesian rose tree with a standard block-model likelihood. So, the innovations are all in how this new likelihood changes the way sufficient statistics are recursively updated in the greedy agglomerative clustering. The likelihood can be computed in O(#Nodes + #Edges) time, which is certainly a nice feature. Unfortunately a description of this is not very good - there's an allusion to dynamic programming before (10), but it's hard to understand the details. Also regarding recursive updating of sufficient statistics, there are some equations but not a lot of help in understanding why they're correct.

TERMINOLOGY:
It is questionable whether the "nonparametric" terminology is appropriate here - Blundell et al. [12] phrase the rose tree approach in terms of model selection, and contrast it with nonparametric methods in their discussion. I am in agreement with their interpretation.

EXPERIMENTS:
The qualitative experiment on Sampson is fine, and the comparison to the IRM is nice in terms of showing the benefits of averaging over many hypotheses when making predictions. Again the biggest limitation is the lack of comparison to other hierarchical network models. There is some discussion about scalability, but none of the networks tested here are especially huge.
Summary: Capably integrates prior work on Bayesian rose trees with stochastic block model likelihoods for networks, and generalizes efficient agglomerative clustering methods to this scenario. This allows effective hierarchical community discovery in experiments, but comparisons to other hierarchical network models are lacking.
Author Feedback

Author rebuttal: We thank the reviewers for their time and effort in producing such helpful
and insightful feedback.

Assigned_Reviewer_2:
We agree it would be nice to quantify the optimality of BHCD.
The original BRT paper did an exhaustive search, on sizes up to 8 items, showing
that the tree recovered by BRT is close to the true tree.

We have considered submodularity in both BRT and BHCD, but thus far we have not
found the appropriate representation.

Assigned_Reviewer_4:

We thank the reviewer for noting a number of grammatical mistakes which we shall
correct.

The submission included the posited ``ground truth'' for Sampson's Monastary
data, provided by Sampson, in the first plot of Figure 3, as stated in the
figure caption on line 376. We also replicated this ground truth in the text, on
lines 306 to 308. We agree the role of the first plot of Figure 3 could be
made clearer in the main body of text.

Assigned_Reviewer_6:

We thank the reviewer for making us aware of Koivisto and Sood (2004).

We evaluated BHCD on link prediction on the NIPS data set as ground truth
``true'' communities are not available.
We held out some of the edges (present or absent) and then attempted to
predict whether each test edge was originally present of absent.
This is a common paradigm for evaluation community discovery algorithms:
references [2], [6], [7], [9], [10], [17], [18] all evaluated their algorithms'
predictive ability in this way.
Hence we did not explain our approach as explicitly as we might: this is easily
remedied.

We included qualitative results in Figure 3 on Sampson's Monastary data set,
where a ``ground truth'' is available (and shown in the top plot of Figure 3).

Assigned_Reviewer_7:
We thank the reviewer for noting Ho et al (2012) and Roy et al (2009).
Our emphasis was on producing a computationally efficient approach to modelling
communities.
Both of the above models have complicated and computationally demanding MCMC
inference schemes---which we expect to be slower than that of the IRM, both in
terms of total run time and number of MCMC steps.